# Estimating progress towards meeting women's contraceptive needs in 185 countries: A Bayesian hierarchical modelling study

**Vladimíra Kantorová**\*, **Mark C. Wheldon**, **Philipp Ueffing**, **Aisha N. Z. Dasgupta**

Population Division, Department of Economic and Social Affairs, United Nations, New York, New York, United States of America

\* kantorova@un.org

**Data Availability Statement:** The input data are included in S1 and S2 Data. Data and metadata are contained in files. Data are also available from the online open-access publication: United Nations,

## Abstract

### Background

Expanding access to contraception and ensuring that need for family planning is satisfied are essential for achieving universal access to reproductive healthcare services, as called for in the 2030 Agenda for Sustainable Development. Monitoring progress towards these outcomes is well established for women of reproductive age (15–49 years) who are married or in a union (MWRA). For those who are not, limited data and variability in data sources and indicator definitions make monitoring challenging. To our knowledge, this study is the first to provide data and harmonised estimates that enable monitoring for all women of reproductive age (15–49 years) (WRA), including unmarried women (UWRA). We seek to quantify the gaps that remain in meeting family-planning needs among all WRA.

### Methods and findings

In a systematic analysis, we compiled a comprehensive dataset of family-planning indicators among WRA from 1,247 nationally representative surveys. We used a Bayesian hierarchical model with country-specific time trends to estimate these indicators, with 95% uncertainty intervals (UIs), for 185 countries. We produced estimates from 1990 to 2019 and projections from 2019 to 2030 of contraceptive prevalence and unmet need for family planning among MWRA, UWRA, and all WRA, taking into account the changing proportions that were married or in a union. The model accounted for differences in the prevalence of sexual activity among UWRA across countries.

Among 1.9 billion WRA in 2019, 1.11 billion (95% UI 1.07–1.16) have need for family planning; of those, 842 million (95% UI 800–893) use modern contraception, and 270 million (95% UI 246–301) have unmet need for modern methods. Globally, UWRA represented 15.7% (95% UI 13.4%–19.4%) of all modern contraceptive users and 16.0% (95% UI 12.9%–22.1%) of women with unmet need for modern methods in 2019. The proportion of the need for family planning satisfied by modern methods, Sustainable Development Goals (SDG) indicator 3.7.1, was 75.7% (95% UI 73.2%–78.0%) globally, yet less than half of the need for family

Department of Economic and Social Affairs, Population Division. World Contraceptive Use 2019. New York: 2019. Available from: https://www.un.org/en/development/desa/population/.

**Funding:** VK, MCW, PU, and ANZD received funding from the Bill and Melinda Gates Foundation at https://www.gatesfoundation.org/ (Grants Nos. OPP1183453 and OPP1110679). The funders had no role in study design, data collection and analysis, decision to publish, or preparation of the manuscript.

**Competing interests:** The authors have declared that no competing interests exist.

**Abbreviations:** DHS, Demographic and Health Surveys; EWEC, Every Woman Every Child; FP2020, Family Planning 2020; GATHER, Guidelines for Accurate and Transparent Health Estimates Reporting; GGS, Gender and Generations Survey; ICPD, International Conference on Population and Development; IUD, intrauterine device; LAM, lactational amenorrhoea method; MCMC, Markov chain Monte Carlo; MICS, Multiple Indicator Cluster Surveys; MWRA, women of reproductive age (15–49 years) who are married or in a union; PMA2020, Performance, Monitoring and Accountability 2020; PPD, posterior probability of a decrease; PPI, posterior probability of an increase; SDG, Sustainable Development Goals; UI, uncertainty interval; UWRA, women of reproductive age (15–49 years) who are unmarried and not in a union; WRA, women of reproductive age (15–49 years).

planning was met in Middle and Western Africa. Projections to 2030 indicate an increase in the number of women with need for family planning to 1.19 billion (95% UI 1.13–1.26) and in the number of women using modern contraception to 918 million (95% UI 840–1,001).

The main limitations of the study are as follows: (i) the uncertainty surrounding estimates for countries with little or no data is large; and (ii) although some adjustments were made, underreporting of contraceptive use and needs is likely, especially among UWRA.

## Conclusions

In this study, we observed that large gaps remain in meeting family-planning needs. The projected increase in the number of women with need for family planning will create challenges to expand family-planning services fast enough to fulfil the growing need. Monitoring of family-planning indicators for all women, not just MWRA, is essential for accurately monitoring progress towards universal access to sexual and reproductive healthcare services—including family planning—by 2030 in the SDG era with its emphasis on 'leaving no one behind.'

## Author summary

### Why was this study done?

- Increasing availability of modern contraception has permitted greater opportunities for individual choice and responsible decision-making in matters of reproduction.
- Global, regional, and national annual estimates of contraceptive prevalence, unmet need for family planning, and the proportion of need for family planning satisfied by modern methods have previously been available only for married or in-union women of reproductive age (15–49 years) (MWRA).
- We sought to provide estimates of these family-planning indicators for all women of reproductive age (15–49 years) (WRA), including unmarried women (UWRA), that could be used for global monitoring of the proportion of WRA who have their need for family planning satisfied with modern methods (Sustainable Development Goals [SDG] indicator 3.7.1.)

### What did the researchers do and find?

- We compiled data on contraceptive use and needs from 1,247 nationally representative surveys and produced harmonised annual estimates and projections that allow monitoring for all WRA, including UWRA.
- Among the 1.9 billion WRA worldwide in 2019, 1.1 billion have demand for family planning; of these, 842 million are using modern contraception, and 270 million have an unmet need for modern methods.
- UWRA account for an increasing share of users of modern contraceptive methods globally (15.7% in 2019 up from 12.1% in 2000), driven in part by an increase in

contraceptive use among UWRA and an increase in the proportion of women who are unmarried.

- In 43 countries, of which 32 are low-income countries, less than half of the need for family planning is met by modern methods.

## What do these findings mean?

- Annual estimates of key family-planning indicators among all women and by marital status, including the degree of uncertainty around them, provide the global health and development community with better tools for monitoring progress in the era of the SDG and other family-planning initiatives.

- Although contraceptive use has increased, the progress has been uneven between countries, and large gaps remain in meeting family-planning needs. These gaps should be met by family-planning programmes that enable couples and individuals to decide freely and responsibly the number and spacing of their children, as well as have the information and means to do so.

## Introduction

Contraception is one of the most important tools that women and men have for achieving desired family size. The Programme of Action of the International Conference on Population and Development (ICPD), adopted in Cairo, Egypt, by 179 governments in 1994, recognised the basic right of all couples and individuals to decide freely and responsibly the number, spacing, and timing of their children and to have the information and means to do so, as well as the right to attain the highest standard of sexual and reproductive health [1]. Past estimates of family-planning indicators have been largely limited to married or in-union women of reproductive age (15–49 years) (MWRA), and unmarried women of reproductive age (15–49 years) (UWRA) have not been paid adequate attention. Recent changes in the international family-planning field have refocused attention towards all women of reproductive age (15–49 years) (WRA), regardless of marital status. Three ongoing international initiatives reaffirming the commitments made in the Programme of Action—the 2030 Agenda for Sustainable Development, Every Woman Every Child (EWEC), and Family Planning 2020 (FP2020)—demand robust data collection, analysis, and monitoring of family-planning indicators among all WRA, regardless of marital status, to track progress towards expanding access to family planning, evaluate programmatic efforts, and identify funding gaps [2–4]. The 2030 Agenda— anchored around 17 Sustainable Development Goals (SDG)—includes under Goal 3, target 3.7 to 'ensure universal access to sexual and reproductive healthcare services, including family planning, by 2030.' EWEC aims to end preventable deaths of women, children, and adolescents and to ensure their health and well-being, which requires universal access to sexual and reproductive healthcare services (including family planning) and rights by 2030. FP2020 aims to expand access to family planning to an additional 120 million women and girls in 69 of the world's poorest countries by 2020, compared to a baseline in 2012.

Robust analysis of family-planning levels and trends among all WRA is impeded by limited data availability. For the population of women who are married or in a union, this limitation has been addressed by producing annually revised data compilations and model-based estimates and projections of family-planning indicators [5–9]. Efforts to estimate and project family-planning indicators for UWRA or for all WRA have not yet accounted for considerable cross-country variations in reproductive behaviour and sexual activity among unmarried women, nor have they captured changes, by age over time, in the proportions of women who are married or in union. Previous work in this area has been based on a limited number of data sources, focused on selected areas of the world, provided estimates for one point in time, restricted to MWRA, or limited the study to one family-planning indicator without understanding the relationship between need for—and use of—contraception over time and across countries [10–16].

We present here a new data compilation and method to estimate indicators of contraceptive prevalence and unmet need for family planning among all WRA, including UWRA. The method expands a Bayesian hierarchical model developed for MWRA [6,17,18] to produce estimates and projections for UWRA and accounts for differences in the prevalence of sexual activity among UWRA. By combining the UWRA results with those for MWRA, estimates and projections were produced for all WRA while incorporating the underlying compositional changes over time in age structures and the proportions of women who are married or in a union.

## Methods

### Definitions

In this study, contraceptive prevalence was defined as the percentage of women who report themselves or their partners as currently using at least one contraceptive method of any type. For analytical purposes, contraceptive methods are classified as either modern or traditional. Modern methods of contraception include female and male sterilization, the intrauterine device (IUD), the implant, injectables, oral contraceptive pills, male and female condoms, vaginal barrier methods (including the diaphragm, cervical cap, and spermicidal foam, jelly, cream, and sponge), the lactational amenorrhoea method (LAM), emergency contraception, and other modern methods not reported separately (e.g., the contraceptive patch or vaginal ring). Traditional methods of contraception include rhythm (e.g., fertility awareness-based methods, periodic abstinence), withdrawal, and other traditional methods not reported separately. Unmet need for family planning is the percentage of women who want to stop or delay childbearing for at least 2 years but are not using any contraceptive method. Unmet-need calculation is based on the definition and computation used by the Demographic and Health Surveys (DHS) [19] (Section 2.2 in S1 Appendix). Demand for family planning satisfied by modern methods (SDG indicator 3.7.1 'Proportion of women who have their need for family planning satisfied with modern methods') is modern contraceptive prevalence divided by the sum of contraceptive prevalence and unmet need (also referred to as total demand for family planning). We provide estimates for all WRA, UWRA, and MWRA.

MWRA pertains to women who are married (defined in relation to the marriage laws or customs of a country) and to women in a union, which refers to women living with their partner in the same household (also referred to as cohabiting unions, consensual unions, unmarried unions, or 'living together'). UWRA pertains to women who are not married and not in a union and is a complement to MWRA. To the extent possible, this distinction was applied to all survey-based observations, or a bias was assigned to data points that had different definitions of marital or union status.

While family-planning indicators among MWRA are commonly used and readily inter-preted, more consideration needs to be given to the calculation and interpretation of the fam-ily-planning indicators among UWRA and all WRA.

The major differences in the approaches used for estimating family-planning indicators among UWRA were related to whether the universe of the population was all UWRA or only UWRA who are deemed to be sexually active (defined by sexual activity in past 28 days, past 3 months, past 1 year, or ever sexually active) and so potentially exposed to pregnancy. In this paper, we report indicators for the population of all UWRA. Contraceptive users among UWRA refers to all UWRA who report in a survey using a contraceptive method, irrespective of sexual activity. For unmet need among all UWRA, it is necessary to determine the timing of their most recent sexual activity. UWRA who are not pregnant or postpartum amenorrhoeic are considered currently at risk of pregnancy (and thus could potentially be included in the numerator as having unmet need) if they have had intercourse within the 4 weeks prior to the survey interview. Unmet need for UWRA who are pregnant or postpartum amenorrhoeic is determined regardless of their most recent sexual activity either as pregnant UWRA whose pregnancies were unwanted or mistimed at the time of conception or postpartum amenor-rhoeic UWRA who are not using family planning and whose last birth was unwanted or mis-timed. Using this approach, estimates of family-planning indicators for all women are directly comparable to those published generally in the survey reports.

## Data compilation

The data compilation uses data from multiple sources and employs unified definition and computation of survey-based estimates of family-planning indicators for UWRA. We reviewed all nationally representative household surveys that provided information on contraceptive prevalence. The starting point was the data sources used for estimating contraceptive use for MWRA [18], complemented by queries to national statistical offices, research institutes, and international survey programs. For each survey, we determined whether the question on cur-rent use of contraception was asked among all women and whether it was possible to estimate contraceptive prevalence and unmet need for family planning among UWRA. The data com-pilation includes 1,247 observations of contraceptive prevalence for 195 countries or areas for the period from 1950 to 2019, 540 observations of unmet need for family planning for MWRA across 143 countries, 551 observations across 136 countries or areas for contraceptive preva-lence, and 250 observations across 72 countries or areas for unmet need for UWRA (Section 2.3 in S1 Appendix) [5]. For 361 surveys—including DHS; Multiple Indicator Cluster Surveys (MICS); Performance, Monitoring and Accountability 2020 (PMA2020) surveys, Gender and Generations Survey (GGS); and other surveys—we obtained estimates, including sampling errors, from micro-datasets. For other surveys, we used estimates derived from published tabu-lations, or we obtained specific tabulations from the institutions responsible for data collec-tion. Many of these survey-based estimates had not been previously published. The input data are provided as Supporting Information (S1, S2 and S3 Data).

Ethics approval was not required for this study.

## Statistical model

For MWRA, we used existing statistical models [8,18] on the updated data compilation. We expanded the model to UWRA (see Section 3 in S1 Appendix). Both models were fitted to the data described earlier, and we report the model-based estimates and projections. The model provides an assessment of uncertainty in the estimates based on the availability and quality of input data. It allows for greater precision when more and better data are available and indicates

the extent of uncertainty in cases in which the data are insufficient or are from sources more susceptible to systematic differences as captured by the biases assigned to each survey-based data point. We describe some key aspects of it subsequently; for a full description, see Section 3 in S1 Appendix. For comparisons between model outputs and the underlying data obtained from surveys for each country, see S2 Appendix.

For each country, the systematic trends over time in total contraceptive prevalence and the ratio of modern-method use to use of any method were modelled with logistic growth curves. The logistic function is appropriate for representing social diffusion processes such as the adoption of contraceptive methods, such that the pace of adoption is slow at first and then is expected to increase at a faster rate, after which it slows down once prevalence approaches higher levels [20,21]. Autocorrelated error processes were added to account for deviations, including reversals, from these smooth trends. Unmet need among UWRA was similarly modelled as a systematic trend plus autocorrelated error but as a function of total contraceptive prevalence rather than time. This function also captures the trend in countries with low sexual activity among UWRA where both contraceptive prevalence and unmet need are very low.

Country-specific estimates of model parameters were obtained by fitting a Bayesian hierarchical model [22], which is fully described in Section 3.6 in S1 Appendix. The output of a Bayesian model is a 'posterior' distribution, a high-dimensional probability distribution for the parameters of interest. It is the result of combining information about the parameters contained in the data (encapsulated in the 'likelihood' component of the model) with existing prior knowledge (encapsulated by the 'prior distribution' component). Standard practice is to summarise the posterior distribution with quantiles of key parameters that provide point estimates and ranges representing the magnitude of uncertainty. We report posterior medians for point estimates and uncertainty intervals (UIs) constructed from the 2.5th and 97.5th percentiles. Under the model, and given the available data, there is a 95% probability that these UIs contain the true value of the parameter.

Many countries had only a few data points, and some countries had none. For example, 59 countries with data on contraceptive prevalence for MWRA had no such data for UWRA. Of the 136 countries that did have data, only 26% of them had any data before 1990 (compared with 63% for MWRA). To improve the precision and accuracy of estimates and projections, a hierarchical structure was added to the Bayesian model. Hierarchical structures cluster countries together based on common characteristics and allow 'borrowing of strength' among observations within the same cluster. Estimates for countries with little, biased, or no data are based on data for other countries in the cluster—partly if they have some data, entirely if they have none. The model implicitly weights data from other countries based on the hierarchical structure positions of the countries.

The model for MWRA used a purely geographic clustering [18]; in other words, countries geographically near each other were clustered together. To improve estimation for UWRA in countries with very few—or no—data points and to account for the cross-country variations in reproductive behaviour and sexual activity among UWRA that have an impact on family-planning indicators, we developed a 2-category sexual activity classification and combined it with geographical clusters. The 2 categories were (i) countries with very low levels of sexual activity (Group 0) and (ii) all other countries (Group 1). Eighty-one countries were assigned using information about proportion sexually active from the most recent DHS or MICS; countries with less than 2% sexually active (defined as having sexual intercourse in the past 28 days) among UWRA were assigned to Group 0 and the rest to Group 1. An additional 43 countries were classified using information about the acceptance of sex between unmarried adults reported in the Pew 2013 Global Attitudes Survey (29 countries) or the World Values Survey Wave 6 (14 countries), and the remaining 71 countries (all in Asia and Northern Africa) were

assigned on the basis of the proportion religious in the population (Section 2.4 and Fig D in S1 Appendix).

Estimates for countries and indicators with no data were obtained from weighted averages of parameter estimates for certain sets of countries with data. The sets and weights were determined by the hierarchical structure of the model, such that averages based on countries most proximate in the hierarchy were given higher weights (Sections 3.4 and 3.7 in S1 Appendix). We included additional parameters in the model to account for surveys that used different target populations (e.g., nonstandard age groups, omission of subnational geographies) or a different categorization of contraceptive method use (the bias and misclassification parameters are described in Section 3.5 in S1 Appendix). For instance, some surveys include women of different age groups compared to the baseline population of women aged 15 to 49 years (e.g., the GGS, which surveyed only women aged 18 and older). To account for differences in data quality, we estimated total errors by data source type.

Estimates for years outside the periods of data availability were obtained from the fitted Bayesian model by interpolation and extrapolation of the systematic logistic trends and the autocorrelated error processes. The parameters of the systematic trend component are time invariant (Section 3.3 in S1 Appendix) and thus provide estimates of the trend for all years. The autocorrelated error process is also parameterised by time-invariant parameters. Once these were obtained from the model fit, extrapolations beyond the period of data availability were obtained by sequentially sampling from the error process conditional distributions. We labelled all estimates for years 2020 and beyond as 'projections.'

We used a Markov chain Monte Carlo (MCMC) algorithm to generate samples from the joint posterior distribution of model parameters [22,23,24]. The MCMC sampling algorithm was implemented using JAGS 4.2.0 Open Source software [24], and the analysis was carried out in R version 3.5.2 [25]. The source code is available at https://github.com/FPcounts/FPEMglobal.

The samples from the posterior were transformed to produce probabilistic results for other indicators, such as need for family planning satisfied by modern methods. Results on the count scale, such as the number of contraceptive users or women with unmet need, were produced by multiplying posterior sample elements by country-specific estimates and projections of the number of UWRA or MWRA, as appropriate [26,27]. For instance, the posterior distribution of the number of UWRA using any method of contraception was obtained by multiplying each element of the posterior sample for contraceptive prevalence by the number of UWRA in the appropriate year, within country. Estimates for country aggregates (e.g., sub-Saharan Africa) were obtained by summing the posterior distributions of the counts across the countries in the aggregate. Estimates for UWRA and MWRA were combined to provide WRA estimates in the same way. That is, for each country at each year, UWRA and MWRA estimates were summed on the count scale to produce WRA estimates.

A key contribution of this article is the data compilation and statistical model for estimating contraceptive prevalence and unmet need for family planning among UWRA itself, therefore a prespecified analysis plan was not prepared, and all analyses are non-prespecified. The statistical model for MWRA [8,18] is applied to a larger and more recent dataset. As in Alkema and colleagues' work [18], model development began with substantive reasoning about the prevalence of contraceptive use and needs and how they might differ between MWRA and UWRA. We identified sexual activity among UWRA as a major factor to be included in the hierarchical structure of the model for UWRA.

The validity of the model was assessed with 3 cross-validation exercises [22]. These entailed leaving out certain sets of observations (depending on the exercise), refitting the model, and checking model results against the left-out observations for calibration and predictive accuracy. The results indicated that the model was robust and adequately calibrated to generate the

estimates for family-planning indicators among UWRA (Section 4.3 in S1 Appendix). We also tested the sexual activity classification for UWRA. Because there were 2 levels ('low' and 'other'), we identified countries with data that were close to the threshold of 2% sexually active among UWRA. The country closest to the threshold was Indonesia, which was assigned to the 'low' sexual activity group. A sensitivity analysis was conducted, assigning Indonesia to the 'other' group, but its relatively abundant data resulted in no meaningful change in the results.

### Indicators used

We computed the medians and 95% UIs for all indicators of interest using the 2.5th and 97.5th percentiles of the posterior distributions for the following 7 indicators: contraceptive prevalence (any, modern, traditional), unmet need for family planning, unmet need for modern methods, total demand for family planning, and demand for family planning satisfied with modern methods.

We generated posterior estimates of change in family-planning indicators over periods of years at the trajectory level so as to obtain probabilistic estimates of change with posterior probabilities of an increase (PPIs) or decrease (PPDs; such that PPD = 1 − PPI). These probabilities reflect the certainty regarding the reported change: a higher posterior probability corresponds to greater certainty about the result. For example, PPI > 0.9 means that there is more than 90% probability that the actual change was greater than 0. These posterior estimates of change and PPIs account for correlation between estimates of the start and end years and are more accurate than, for example, comparing the separately generated posterior estimates (including uncertainty) at the start and end years of the period.

We produced national, regional, and global results for the period from 1990 to 2030 and discuss the changes in estimates from 2000 to 2019 and projections to 2030 in line with the requirements for SDG monitoring. Results quoted for 2030 are model-based predictions, and we use different visual elements to distinguish these in plots. We present results for the 185 countries with at least 90,000 inhabitants in 2017 [26]. Estimates from an additional 10 countries and areas with total population below 90,000 inhabitants are included in the aggregated results. The classifications used to form regions, subregions, and other country aggregates are defined in S1 Appendix Table G. The results are reported according to the Guidelines for Accurate and Transparent Health Estimates Reporting (GATHER) statement [28] (S1 Checklist).

## Results

### Family-planning trends among MWRA

Modern contraceptive prevalence among MWRA increased worldwide between 2000 and 2019 from 55.0% (95% UI 53.7%–56.3%) to 57.1% (95% UI 54.6%–59.5%), or 2.1 percentage points (95% UI −0.7 to 4.8, PPI > 0.93) (Table 1 and Fig 1; Section 5 in S1 Appendix, S4 and S5 Results for country-specific results; Table 2 for corresponding numbers of women). Prevalence in 2019 was highest in Eastern Asia (80.8% [95% UI 74.8%–85.4%]). Median estimates for Western Europe, South America, and Northern Europe were above 70%, while those for Northern America, Central America, and Australia and New Zealand were above 60%. In contrast, prevalence was lowest in 2019 in Middle Africa (13.7% [95% UI 10.5%–17.9%]) and Western Africa (20% [95% UI 17.8%–22.5%]). These were, however, increases over the 2000 levels by, respectively, 7.1 percentage points (95% UI 3.5–11.5) and 11.7 percentage points (95% UI 9.3–14.4), with PPI = 1 in both cases. The increases were also large in Eastern Africa, greater by 23.7 percentage points (95% UI 20.9–26.7, PPI = 1), and Eastern Europe, greater by 10.5 percentage points (95% UI 0.6–19.9, PPI > 0.98).

**Table 1. Global estimates of the proportion (%) of WRA using a modern contraceptive method, having unmet need for modern methods, and having demand for family planning, and the proportion of need for family planning satisfied by modern methods, for 1990, 2000, 2010, 2019, and 2030.**

| | | All WRA (%) | MWRA (%) | UWRA (%) |
|---|---|---|---|---|
| **Indicator** | **Year** | **Median (95% UI)** | **Median (95% UI)** | **Median (95% UI)** |
| **Contraceptive use (any modern method)** | 1990 | 35.9 (34.0–39.3) | 47.5 (45.1–49.6) | 12.3 (8.9–22.0) |
| | 2000 | 42.0 (40.5–44.6) | 55.0 (53.7–56.3) | 15.4 (12.0–23.1) |
| | 2010 | 43.5 (41.6–46.4) | 56.7 (54.7–58.6) | 18.3 (14.5–25.8) |
| | 2019 | 44.3 (42.1–47.0) | 57.1 (54.6–59.5) | 20.1 (16.3–26.3) |
| | 2030 | 45.1 (41.2–49.1) | 59.2 (54.3–63.7) | 21.5 (17.1–27.8) |
| **Contraceptive use (any traditional method)** | 1990 | 6.1 (5.3–8.0) | 7.7 (6.8–8.6) | 3.0 (1.4–8.3) |
| | 2000 | 5.2 (4.6–6.3) | 6.6 (5.9–7.3) | 2.4 (1.4–5.6) |
| | 2010 | 4.5 (3.9–5.5) | 5.9 (5.2–6.9) | 1.9 (1.1–4.3) |
| | 2019 | 4.2 (3.5–5.2) | 5.6 (4.6–6.9) | 1.6 (1.0–3.3) |
| | 2030 | 3.9 (3.0–5.3) | 5.4 (4.0–7.4) | 1.5 (0.9–3.0) |
| **Unmet need for modern methods** | 1990 | 17.6 (16.3–20.1) | 22.7 (21.3–24.2) | 7.3 (4.6–14.3) |
| | 2000 | 15.1 (14.1–16.8) | 19.1 (18.2–20.0) | 7.0 (4.8–11.9) |
| | 2010 | 14.3 (13.2–15.9) | 18.2 (17.1–19.6) | 6.7 (4.7–11.0) |
| | 2019 | 14.2 (12.9–15.8) | 18.2 (16.7–20.0) | 6.6 (4.8–10.1) |
| | 2030 | 13.3 (11.6–15.5) | 17.3 (14.9–20.3) | 6.6 (4.9–9.9) |
| **Total need for family planning** | 1990 | 53.6 (51.5–57.6) | 70.2 (68.4–71.8) | 19.6 (14.6–31.2) |
| | 2000 | 57.1 (55.4–60.0) | 74.1 (73.0–75.1) | 22.4 (17.7–31.1) |
| | 2010 | 57.8 (55.8–60.7) | 74.9 (73.4–76.4) | 25.0 (20.1–33.1) |
| | 2019 | 58.5 (56.5–61.1) | 75.3 (73.6–77.0) | 26.7 (22.1–33.2) |
| | 2030 | 58.4 (55.3–61.9) | 76.5 (73.3–79.3) | 28.1 (23.0–34.7) |
| **Proportion of the need for family planning satisfied by modern methods (SDG indicator 3.7.1)** | 1990 | 67.1 (63.8–69.7) | 67.7 (65.4–69.7) | 62.7 (45.6–76.4) |
| | 2000 | 73.5 (71.3–75.3) | 74.3 (73.0–75.4) | 68.6 (56.2–78.6) |
| | 2010 | 75.3 (72.9–77.3) | 75.7 (73.8–77.3) | 73.2 (61.9–81.7) |
| | 2019 | 75.7 (73.2–78.0) | 75.8 (73.4–78.0) | 75.4 (66.1–82.6) |
| | 2030 | 77.2 (73.2–80.5) | 77.4 (73.1–80.9) | 76.6 (67.7–83.4) |

**Abbreviations**: MWRA, women of reproductive age (15–49 years) who are married or in a union; SDG, Sustainable Development Goals; UI, uncertainty interval; UWRA, women of reproductive age (15–49 years) who are unmarried and not in a union; WRA, women of reproductive age (15–49 years)

Over the same period, the proportion of the need for family planning satisfied by modern methods among MWRA worldwide changed from 74.3% (95% UI 73.0%–75.4%) to 75.8% (95% UI 73.4%–78.0%) (Table 1 and Fig 2; Table 2 for corresponding numbers of women). The median estimate of change was 1.5% (95% UI −1% to 3.9%, PPI = 0.88). Eastern Asia and Western Europe had the highest proportions of need satisfied (92.9% [95% UI 88.9%–95.6%] and 89.7% [95% UI 83.5%–93.6%], respectively). Middle Africa (27.3% [95% UI 21.9%–33.6%]) and Western Africa (42.3% [95% UI 38.2%–46.6%]) had the lowest estimated proportions of need satisfied, although these were increases on 2000 levels —as for modern contraceptive prevalence—of 12.4 percentage points (95% UI 6.2–19.4, PPI = 1) and 20.1 percentage points (95% UI 15.3–24.9, PPI = 1), respectively. Eastern Africa had the third lowest estimated proportions of need satisfied in 2000 (32.7% [95% UI 31.1%–34.4%]) but a substantial increase of 29.8 percentage points (95% UI 26.3%—33.3%, PPI = 1) in the 2000–2019 period, which resulted in an estimate of 62.5% (95% UI 59.3%–65.6%) in 2019.

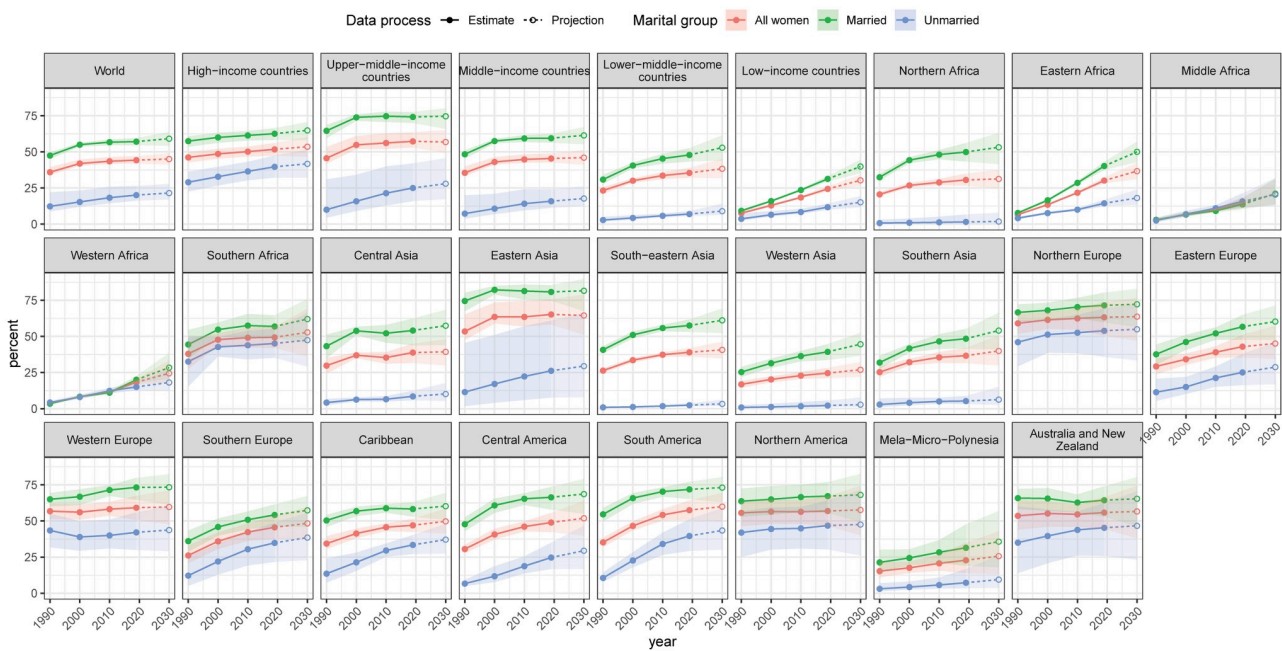

**Fig 1. Percentage of WRA who used a modern contraceptive method by marital group for the period 1990–2030, by World Bank income group and subregion.** The country classification by income level is based on June 2019 gross national income per capita from the World Bank. Estimates are indicated for the years 1990, 2000, 2010, and 2019 by solid circles and interpolated by solid lines. Projections for the year 2030 are indicated by open circles and interpolated by dashed lines. Shaded ribbons indicate 95% UIs; intervals for projection years are shaded lighter. UI, uncertainty interval; WRA, women of reproductive age (15–49 years).

## Family-planning trends among UWRA

Between 2000 and 2019, the proportion of UWRA using modern contraceptive methods increased worldwide from 15.4% (95% UI 12.0%–23.1%) to 20.1% (95% UI 16.3%–26.3%), or 4.6 percentage points (95% UI 0.2–8.7, PPI > 0.98) (see Section 5 in S1 Appendix, S2 and S3 Results for country-specific results). Contraceptive prevalence and the scale of its increase varied widely across geographic subregions. In 2019, more than a third of UWRA were using a modern method in Australia/New Zealand, the Caribbean, Northern America, Northern Europe, South America, Southern Africa, Southern Europe, and Western Europe (Table 1 and Fig 1; Table 2 for corresponding numbers of women). The 2 subregions with the largest increase since 2000 were Central America and South America, greater by 12.7 percentage points (95% UI 3.0–23.4, PPI > 0.99) and 16.8 points (95% UI 6.3–28.7, PPI > 0.99), respectively (Fig 3). Although UWRA contraceptive use generally remained low in sub-Saharan Africa (except Southern Africa), the proportion of UWRA using a modern contraceptive method increased with posterior probability of 99% in Eastern Africa by 6.7 percentage points (95% UI 4.6–9.1), in Middle Africa by 8.9 percentage points (95% UI 4.0–14.7), and in Western Africa by 7.0 percentage points (95% UI 4.2–10.0).

The proportion of UWRA using a modern contraceptive method remained very low in Northern Africa, South-Eastern Asia, and Western Asia (median estimates for the subregions are less than 3%) (Fig 4A). These are subregions in which overall need for family planning and sexual activity among UWRA is low.

The use of traditional methods is not common among UWRA. Globally, 1.6% (95% UI 1.0%–3.3%) of UWRA used a traditional method in 2019. Middle Africa is an exception (5.4% [95% UI 2.6%–10.3%]) with the highest prevalence of traditional methods in Congo (11.5% [95% UI 3.1%–25.1%]).

**Table 2. Global estimates of the number (millions) of WRA using a modern contraceptive method, having unmet need for modern methods, and having demand for family planning, and the proportion unmarried among all women, for 1990, 2000, 2010, 2019, and 2030.**

| Indicator | Year | All WRA, N (in millions) Median (95% UI) | MWRA, N (in millions) Median (95% UI) | UWRA, N (in millions) Median (95% UI) | Proportion unmarried among all women (%) Median (95% UI) |
|---|---|---|---|---|---|
| **Contraceptive use (any modern method)** | 1990 | 473 (448–518) | 420 (399–438) | 53 (39–95) | 11.2 (8.6–18.4) |
| | 2000 | 660 (637–701) | 580 (566–593) | 80 (62–120) | 12.1 (9.7–17.1) |
| | 2010 | 779 (745–830) | 667 (642–689) | 113 (89–159) | 14.4 (12.0–19.1) |
| | 2019 | 842 (800–893) | 709 (678–740) | 133 (107–174) | 15.7 (13.4–19.4) |
| | 2030 | 918 (840–1,001) | 754 (692–812) | 164 (130–212) | 17.9 (15.5–21.2) |
| **Contraceptive use (any traditional method)** | 1990 | 81 (70–105) | 68 (60–76) | 13 (6–36) | 16.1 (9.0–34.2) |
| | 2000 | 82 (72–99) | 69 (62–77) | 12 (7–29) | 15.0 (9.8–29.3) |
| | 2010 | 81 (70–99) | 70 (61–81) | 11 (7–26) | 14.1 (9.7–26.8) |
| | 2019 | 80 (66–99) | 69 (57–85) | 11 (7–22) | 13.5 (10.0–21.9) |
| | 2030 | 80 (61–109) | 69 (51–95) | 12 (7–23) | 14.5 (11.1–21.0) |
| **Unmet need for modern methods** | 1990 | 232 (215–264) | 200 (188–214) | 32 (20–62) | 13.7 (9.3–23.4) |
| | 2000 | 237 (222–264) | 201 (192–211) | 36 (25–61) | 15.3 (11.3–23.2) |
| | 2010 | 255 (237–285) | 214 (201–230) | 41 (29–68) | 16.1 (12.3–23.8) |
| | 2019 | 270 (246–301) | 226 (207–248) | 43 (32–66) | 16.0 (12.9–22.1) |
| | 2030 | 271 (236–317) | 221 (190–258) | 50 (37–75) | 18.5 (15.7–23.8) |
| **Total need for family planning** | 1990 | 705 (678–758) | 620 (604–635) | 85 (63–135) | 12.0 (9.3–17.8) |
| | 2000 | 897 (870–944) | 781 (770–792) | 116 (92–161) | 12.9 (10.5–17.1) |
| | 2010 | 1,035 (999–1,087) | 881 (863–897) | 154 (123–203) | 14.9 (12.3–18.7) |
| | 2019 | 1,112 (1,073–1,161) | 936 (915–957) | 176 (146–219) | 15.8 (13.6–18.8) |
| | 2030 | 1,189 (1,126–1,261) | 975 (934–1,011) | 214 (175–265) | 18.0 (15.6–21.0) |

**Abbreviations**: MWRA, women of reproductive age (15–49 years) who are married or in a union; UI, uncertainty interval; UWRA, women of reproductive age (15–49 years) who are unmarried and not in a union; WRA, women of reproductive age (15–49 years)

The proportion of UWRA with unmet need for modern methods (including traditional-method users) remained unchanged since 2000 and stood at 6.6% (95% UI 4.8%–10.1%) globally in 2019. High proportions of UWRA continued to experience unmet need in Middle and Western Africa, exceeding 15% in Angola, Benin, Congo, Côte d'Ivoire, Democratic Republic of the Congo, Gabon, Guinea-Bissau, Liberia, Madagascar, and Sierra Leone (Fig 4B).

The proportion of need for family planning satisfied by modern methods among UWRA in 2019 was estimated at 75.4% (95% UI 66.1%–82.6%) globally, increasing from 68.6% (95% UI 56.2%–78.6%) in 2000. It remained low in Middle Africa (48.1% [95% UI 37.5%–57.9%]), Western Africa (58.1% [95% UI 52.0%–63.6%]), and Eastern Africa (63.0% [95% UI 58.4%–67.5%]), though it has increased since 2000 by more than 10 percentage points in all 3 subregions. Estimates of the proportion of need for family planning satisfied by modern methods among UWRA is presented for 140 countries (excluding countries classified as having low sexual activity among unmarried women) and among them, the proportion of need satisfied was below 50% in 11 countries and below 75% in an additional 74 countries in 2019 (Fig 4C, Section 5 in S1 Appendix).

## UWRA contraceptive users in the context of all users

UWRA account for an increasing proportion of modern users (from 12.1% [95% UI 9.7%–17.1%] in 2000 to 15.7% [95% UI 13.4%–19.4%] in 2019; Table 2). In Northern Africa, Southern Asia, South-Eastern Asia, and Western Asia, less than 5% of all modern users were UWRA

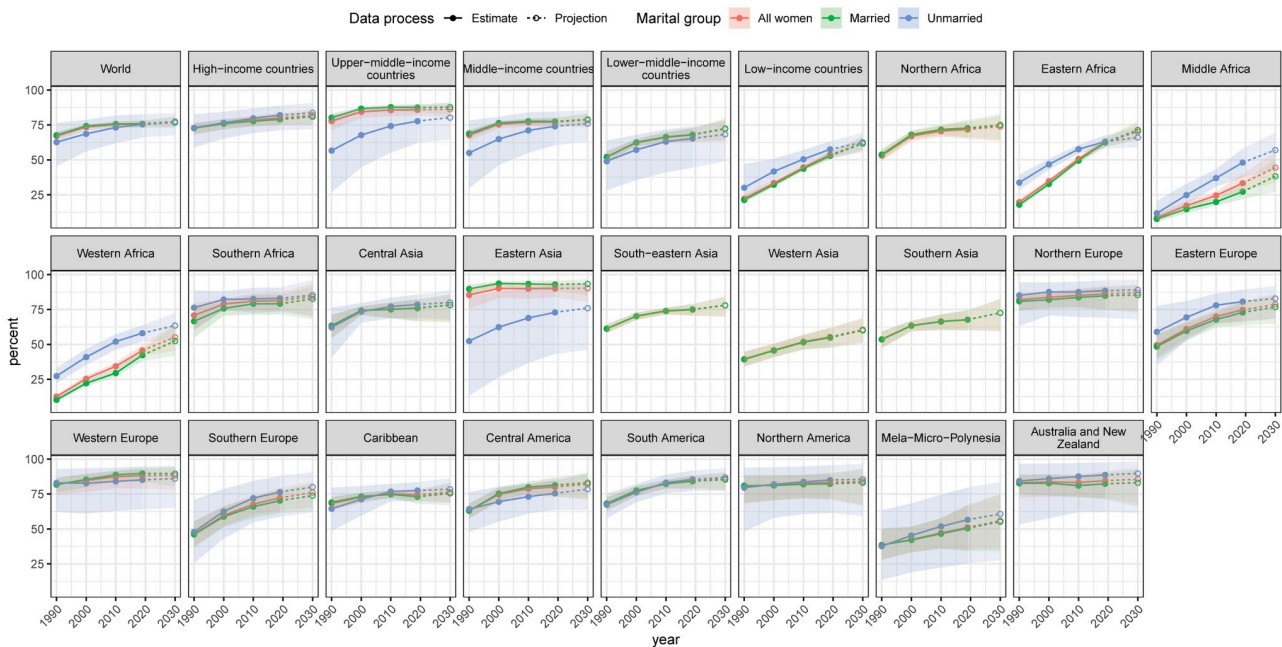

**Fig 2. Proportion of need for family planning satisfied by modern methods among WRA by marital group for the period 1990–2030, by World Bank income group and subregion.** The country classification by income level is based on June 2019 gross national income per capita from the World Bank. Estimates are indicated for the years 1990, 2000, 2010, and 2019 by solid circles and interpolated by solid lines. Projections for the year 2030 are indicated by open circles and interpolated by dashed lines. Shaded ribbons indicate 95% UIs; intervals for projection years are shaded lighter. UI, uncertainty interval; WRA, women of reproductive age (15–49 years).

in 2019, compared to the highest share of 58.5% (95% UI 52.2%–63.7%) in Southern Africa (Fig 5). More than a third of modern contraceptive users are UWRA in Australia/New Zealand, Middle Africa, Northern America, Northern Europe, and Southern Europe. Globally, UWRA accounted for 16.0% (95% UI 12.9%–22.1%) of women with unmet need for modern methods in 2019.

## Contraceptive prevalence and need for family planning among WRA

Among all WRA, use of modern contraceptives increased worldwide between 2000 and 2019 from 42.0% (95% UI 40.5%–44.6%) to 44.3% (95% UI 42.1%–47.0%) (PPI > 0.97) and in absolute number of users from 660 million (95% UI 637–701 million) to 842 million (95% UI 800–893 million) (Tables 1 and 2, Section 5 in S1 Appendix, S6 and S7 Results for country-specific results).

Global unmet need for modern methods declined marginally from 15.1% (95% UI 14.1%–16.8%) in 2000 to 14.2% (95% UI 12.9%–15.8%) in 2019. In absolute numbers, 270 million (95% UI 246–301 million) women have an unmet need for modern methods in 2019, up from 237 million (95% UI 222–264 million) in 2000. The subregion of Southern Asia has the highest number of women with unmet need for modern methods in 2019 at 87 million (95% UI 71–109 million), with 4 other subregions above 15 million: 22 million (95% UI 18–21 million) of Eastern Africa, 20 million (95% UI 18–22 million) of Western Africa, 23 million (95% UI 20–27 million) of South-Eastern Asia, and 28 million (18–48 million) of Eastern Asia (Fig 6).

The proportion of need for family planning satisfied by modern methods among WRA globally increased from 73.5% (95% UI 71.3%–75.3%) in 2000 to 75.7% (95% UI 73.2%–78.0%) in 2019, or 2.2 percentage points (95% UI −0.3 to 4.7, PPI > 0.95) (Fig 7, Table 1). Of

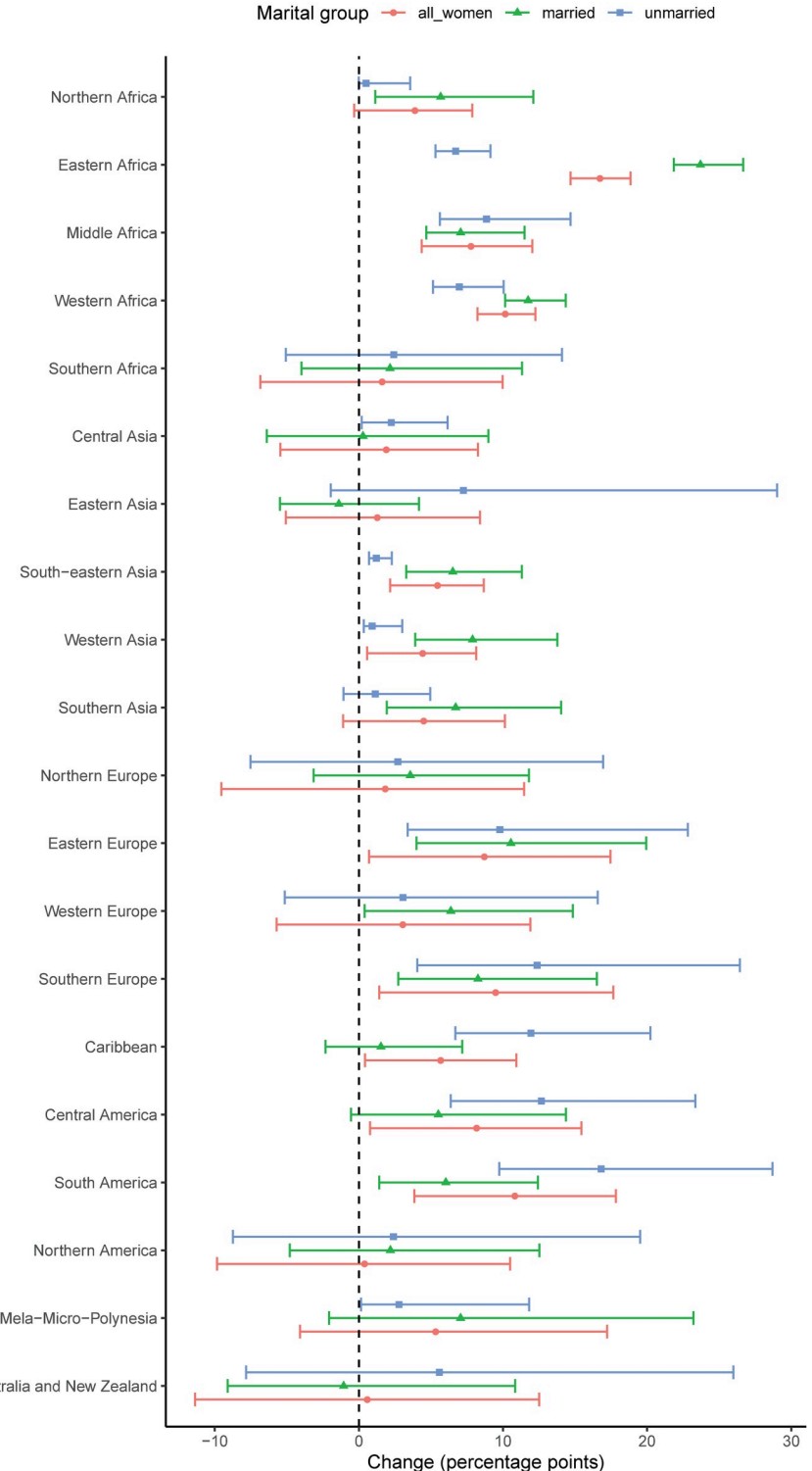

**Fig 3. Percentage-points change between 2000 and 2019 in the proportion of WRA who used a modern contraceptive method, by marital status.** The 95% UIs are displayed by horizontal lines. UI, uncertainty interval; WRA, women of reproductive age (15–49 years).

**A) Contraceptive prevalence of modern methods among unmarried women in 2019 (%)**

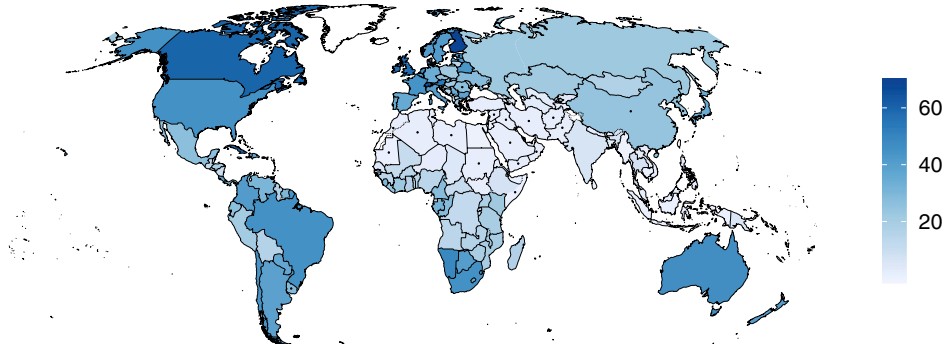

**B) Unmet need for modern methods among unmarried women in 2019 (%)**

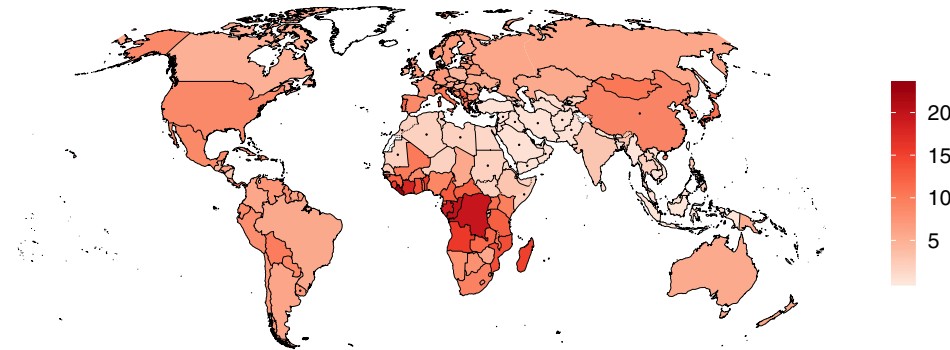

**C) Proportion of need satisfied with modern methods among unmarried women in 2019 (%)**

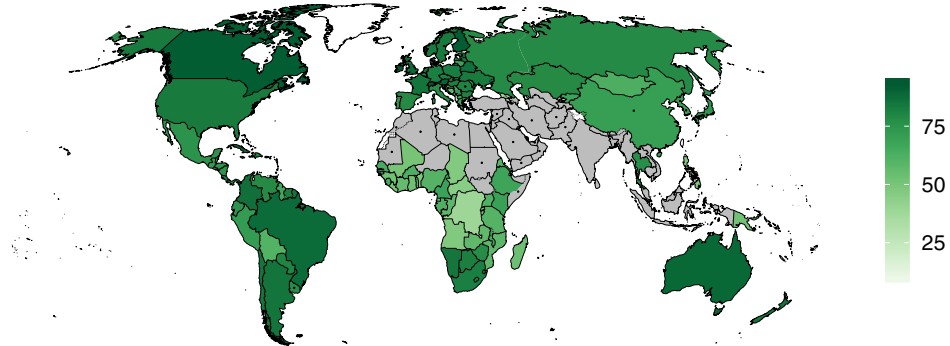

☐ Not reported in countries with low demand for family planning among unmarried women

**Fig 4. 2019 estimates for family-planning indicators among UWRA.** Median estimates of (A) modern contraceptive prevalence, (B) unmet need for a modern method, and (C) the proportion of need for family planning satisfied by modern methods. For countries marked with '*,' no survey-based observations are available for UWRA. The base map was obtained from Natural Earth (https://naturalearthdata.com). The boundaries and names shown and the designations used on this map do not imply official endorsement or acceptance by the United Nations. UWRA, women of reproductive age (15–49 years) who are unmarried and not in a union.

76 countries that in 2000 had less than half of the demand satisfied by modern methods, 34 countries had demand satisfied above 50% in 2019, with the highest increases in Rwanda, from 13.3% (95% UI 11.4%–15.6%) in 2000 to 66.8% (95% UI 56.1%–75.9%) in 2019, and in Ethiopia, from 15.7% (95% UI 13.7%–18.2%) in 2000 to 62.5% (95% UI 55.5%–69.4%) in 2019 (PPI = 1.0) (Fig 8). Our probabilistic projections indicate that countries and regions with

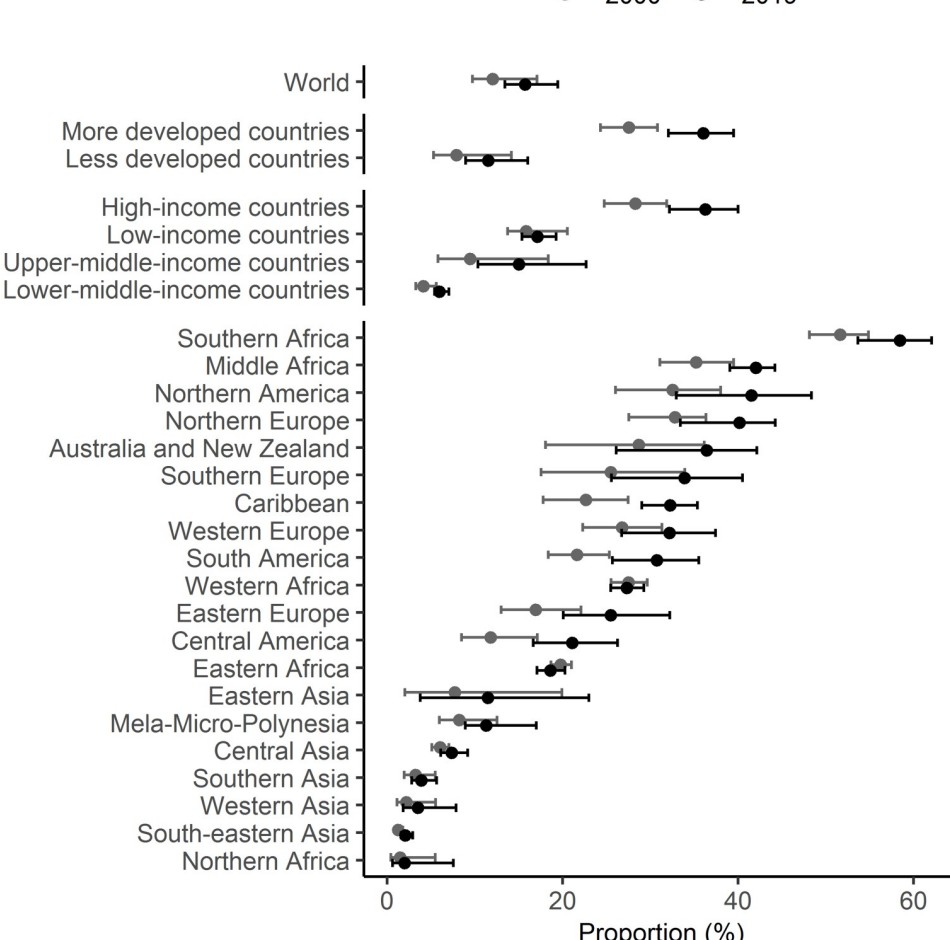

**Fig 5. Proportion of UWRA among WRA using modern contraception by World Bank income group and by subregion in 2000 and 2019.** The 95% UIs are displayed by horizontal lines. In the United Nations classification, the more developed regions comprise all countries of Europe, Northern America, Australia and New Zealand, and Japan. The less developed regions comprise all other countries. The country classification by income level is based on June 2019 gross national income per capita from the World Bank. UI, uncertainty interval; UWRA, women of reproductive age (15–49 years) who are unmarried and not in a union; WRA, women of reproductive age (15–49 years).

currently small proportions of need satisfied would achieve higher levels by 2030 as projected based on country-specific experiences together with experiences in other countries in the region and the world.

The proportion of need satisfied by modern methods remains smallest in countries with low income, despite the large increase from 32.1% (95% UI 30.5%–34.0%) in 2000 to 52.6% (95% UI 50.1%–55.2%) in 2019 (PPI > 0.99) (Fig 7), with the smallest proportions of need satisfied by modern methods in Chad 23.6% (95% UI 15.9%–33.5%), Somalia 17.7% (95% UI 4.6%–44.8%), and South Sudan 17.8% (95% UI 9.5%–35.4%). Among 43 countries with less than half of their demand satisfied by modern methods in 2019, 32 are low and lower-middle income.

## Discussion

Our findings indicate that, among the 1.9 billion WRA worldwide in 2019, 1.1 billion have demand for family planning; of these, 842 million are using modern contraception, and 270

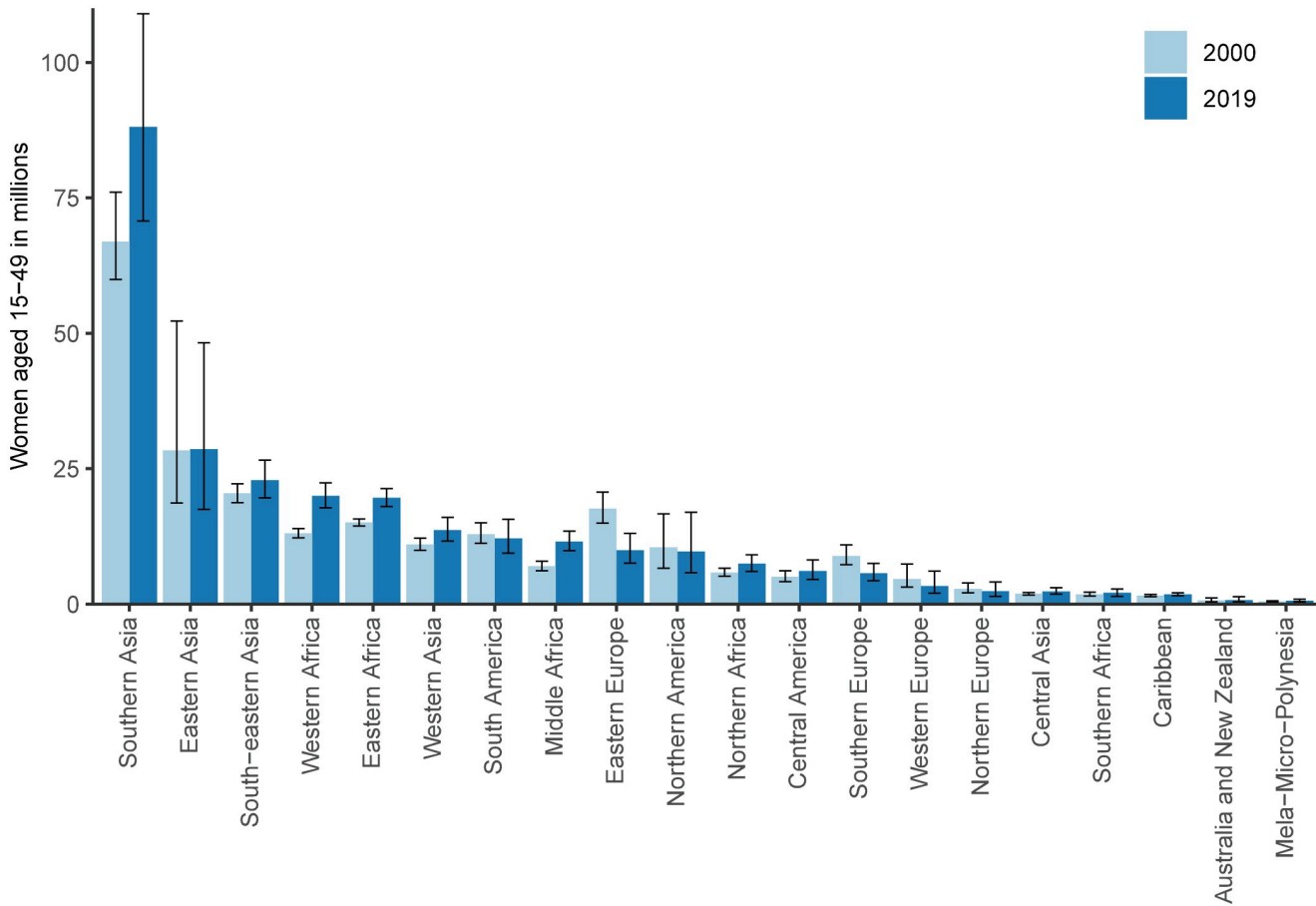

**Fig 6. Number of WRA having unmet need for modern methods by subregion in 2000 and 2019.** The 95% UIs are displayed by vertical lines. UI, uncertainty interval; WRA, women of reproductive age (15–49 years).

million have an unmet need for modern methods. The majority of WRA are in developing countries. In low- and middle-income countries (as defined by the World Bank classification of gross national income per capita as of June 2019), 708 million women are using modern contraception, and 234 million women have an unmet need for modern methods. In less developed regions (as used in the United Nations classification), 698 million women are using modern contraception, and 232 million women have an unmet need for modern methods. In Middle and Western Africa, high proportions of WRA (both among married and unmarried women) continue to experience unmet need for modern methods, highlighting the need for increased government commitment to, and greater international investment in, family-planning programming in these 2 subregions.

The proportion of all modern contraceptive users who are UWRA rose from 12.1% in 2000 to 15.7% in 2019, driven by an increase in modern contraceptive prevalence among UWRA as well as an increasing share of women who are unmarried or not in a union (from 32.9% in 2000 to 34.7% in 2019), the latter due to changes in marital patterns and age structures [26,27]. The contribution of UWRA to the total demand for family planning will likely continue to increase because of further postponement of marriage and union formation; however, it will also depend on the changes in sexual activity among UWRA. A previous study of trends in sexual activity among UWRA [29] observed no generalised pattern over time and across countries.

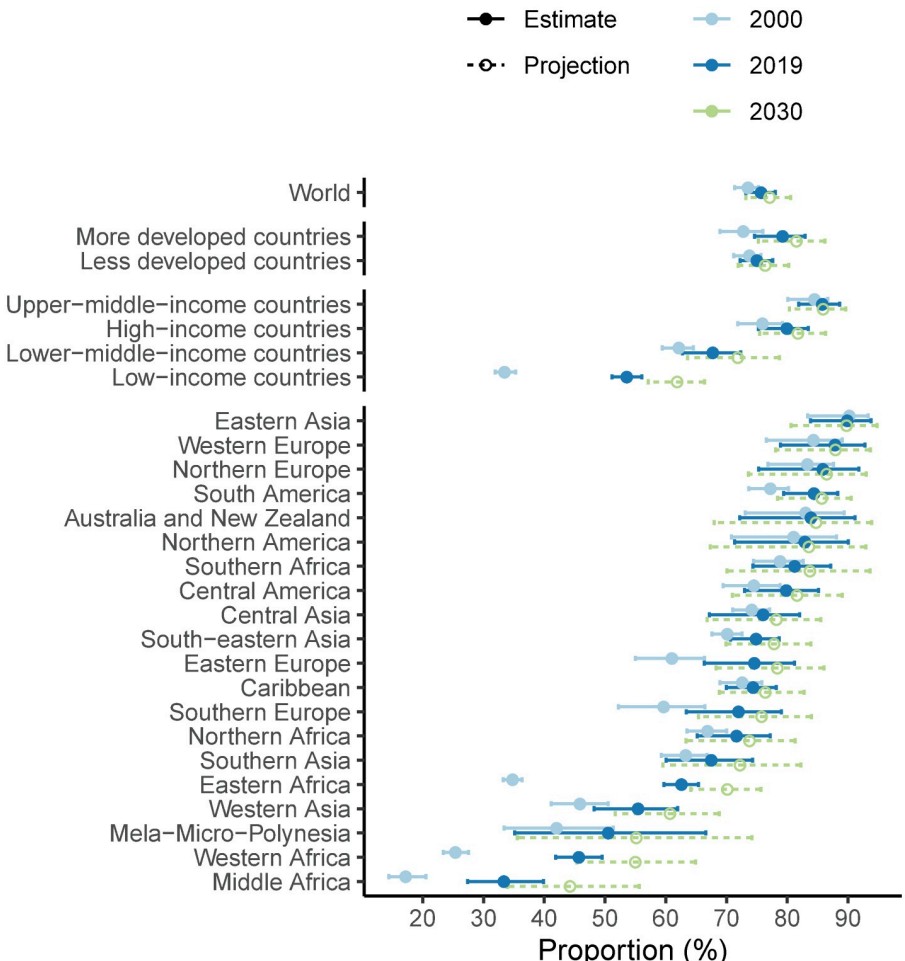

**Fig 7. Proportion of need for family planning satisfied by modern contraceptive methods among WRA, median and 95% UIs, by subregion.** Estimates are shown for the years 2000 and 2019. The 95% UIs are displayed by solid horizontal lines and median estimates by the solid circles. Projections for 2030 are shown using open circles and dashed lines. In United Nations classification, the more developed regions comprise all countries of Europe, Northern America, Australia and New Zealand, and Japan. The less developed regions comprise all other countries. The country classification by income level is based on June 2019 gross national income per capita from the World Bank. Note: results for all countries presented in S2 Appendix. UI, uncertainty interval; WRA, women of reproductive age (15–49 years).

Despite three-quarters of WRA globally having their need for family planning satisfied by modern methods in 2019, there is great diversity across countries and subregions. Most countries with currently low proportions of need for family planning satisfied by modern methods are low-income and lower-middle–income countries, and for many of them, the population of WRA is projected to increase by more than a third by the end of 2030 compared to 2019 [27]. This will create challenges to expand family-planning services fast enough to fulfil the growing need for family planning and will likely generate additional development-related challenges associated with rapid population growth. Even with the projected increases in the proportion of need satisfied by modern methods in each country from 2019 to 2030, the global figure is projected to increase only slightly to 77.2% (95% UI 73.2%–80.5%) in 2030 (Table 1) because of the changing composition of WRA worldwide that reflects increasing representation of women from countries with a lower proportion of need for family planning satisfied with modern methods.

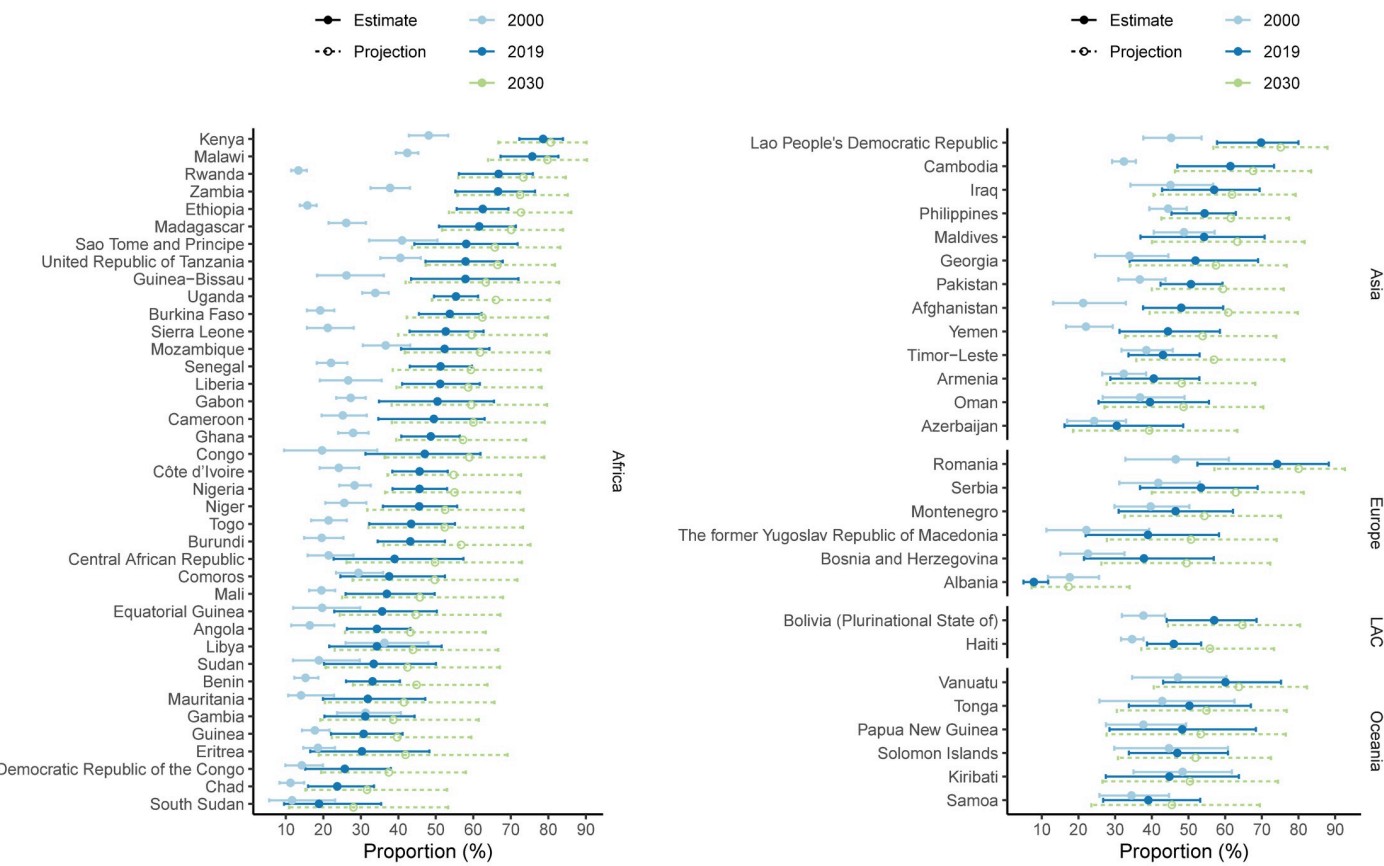

**Fig 8. Proportion of need for family planning satisfied by modern contraceptive methods among WRA, median and 95% UIs, by country, among countries where proportion of need for family planning satisfied by modern contraceptive methods was less than 50% in 2000.** The 95% UIs are displayed by solid horizontal lines and median estimates by the solid circles. Projections for 2030 are shown using open circles and dashed lines. UI, uncertainty interval; WRA, women of reproductive age (15–49 years).

In a comprehensive and systematic way, we generated annual estimates and projections of contraceptive prevalence, unmet need, and demand for family planning among UWRA and all WRA for the period 1990 to 2030. We took into account a wide range of surveys, systematically accounting for variability in errors across data sources, and potential biases in observations that departed from standard measures or reference groups for contraceptive prevalence. The Bayesian approach generated UIs for all estimates and projections, enabled assessments of the probability of actual change over time, and allowed cross-country comparisons and calculation of aggregates for specific time periods. The model accounts for variations in the prevalence of sexual activity among UWRA across countries and incorporates underlying compositional changes in marital/union status over time. For countries or areas that had no data on contraceptive use among UWRA, we provided estimates based on a novel hierarchical classification derived from information on sexual activity among UWRA and geographical clusters.

The most recent comparable study—Adding It Up 2017 by the Guttmacher Institute [13]—estimated 671 million modern contraceptive users among WRA in developing countries for 2017 (compared to our estimate of 685 million [95% UI 655–728 million] for 2017), and they estimated 214 million women with unmet need for modern methods (compared to our estimate of 230 million [95% UI 212–256 million]); both Guttmacher estimates are within our 95% UIs. For sub-Saharan Africa, the differences in the numbers of modern contraceptive

users are larger. The Adding It Up 2017 estimate of 51 million users of modern contraception is lower than the 95% UI of our estimate for year 2017 of 58 million (95% UI 56–60 million), whereas the estimate for unmet need for modern methods is comparable—51 million in Adding It Up compared to our 51 million (95% UI 48–53 million). Adding It Up 2017 did not publish results for individual countries, and therefore we cannot directly compare the results. There are 2 main reasons for discrepancies between the 2 sources. First, more input data have become available describing the situation in 2017. Whereas the most recent surveys in Adding It Up 2017 were from 2016 (and only 2 of the 48 countries of sub-Saharan Africa had a 2016 survey available), our 2019 data compilation contains surveys that took place in 2016 or after for 18 sub-Saharan African countries. Second, the approach of producing current estimates is different. The Adding It Up 2017 study used estimates from surveys in years prior to 2017 as estimates for the situation in 2017. Therefore, in the countries with increasing contraceptive use, the most recent survey-based observation of contraceptive prevalence—especially when time since last survey is several years—might underestimate the current situation. However, our model-based estimates use short-term projections since the last observation, taking into account past trends in the country and regional and global trends.

Nevertheless, some data limitations remain. For 59 of the 195 countries, there are no survey data available for contraceptive use among UWRA (marked with '*' in the maps in Fig 4 and indicated in Tables H–K in S1 Appendix). The models used here provide reasonable estimates of contraceptive prevalence and unmet need for countries without such data, determined by the hierarchical structure of the model, such that averages based on countries most proximate in the hierarchy were given greater weight. However, the uncertainty surrounding those estimates is large. Even for countries where survey data are available, contraceptive use and sexual activity among UWRA are thought to be underreported, and data may be susceptible to other biases. The extent of and methods for adjustment of underreporting require further research. The largest gap in the data compilation for UWRA was China, which represented 14% of the global population of UWRA. In China, only ever-married women were asked about contraceptive use, even in light of growing evidence from studies (though not nationally representative) that sexual activity and contraceptive use among unmarried women is increasingly common (Section 2.3.2 in S1 Appendix). Therefore, the 95% UI around the estimated proportion of UWRA modern contraceptive users is large for Eastern Asia (see Fig 1).

Our priority was to provide estimates for all WRA (comparable to estimates generally published in survey reports), and our study included all contraceptive users irrespective of their sexual activity status. Not all contraceptive users are sexually active. Among MWRA, in many populations more than 10% of women who report currently using a contraceptive method also report no sexual intercourse in the past 28 days. Among UWRA, this proportion was observed to be more than 50% depending on the population and contraceptive method used [29]. Different approaches could have analysed only the population of women who are sexually active and include only those who report being sexually active in the counts of contraceptive users. While a previous study analysed survey data for the population of sexually active WRA only, it did not produce annual estimates and projections or the aggregate results [15]. To produce such aggregates would require estimates of populations of sexually active women across countries and over time.

For estimates of current need for family planning among nonusers (excluding those who are pregnant, postpartum, or infecund), it is commonly assumed that all MWRA are sexually active despite the fact that, in many populations, 20% or more report no sexual intercourse in the past 4 weeks. For UWRA, such an assumption is untenable because there are large differences in the prevalence of recent sexual activity among UWRA [29]; thus, a criterion of sexual exposure to risk of pregnancy is needed. Previous studies have used a variety of criteria: sex in

last 4 weeks, sex in past 3 months, sex in past 1 year, and ever had sex. This has an important implication for the calculation of family-planning indicators among unmarried women, such as unmet need and demand satisfied, in which sexual activity is used to determine exposure to the risk of pregnancy for this group of women. In this paper, the 4-weeks criterion is applied following the unmet need for family planning algorithm in DHS [19]. The definition of current sexual activity as sexual intercourse within the last 4 weeks misses many UWRA who were sexually active within a longer period of time (e.g., within the last year), who may be at risk of pregnancy even if they are only sporadically sexually active.

The estimates presented here will enable monitoring of progress towards universal access to reproductive health and enable assessment of the impact of commitments made and actions taken in the 2030 Agenda under SDG indicator 3.7.1, as well as FP2020 reporting of the number of additional users, contraceptive prevalence, unmet need, and demand for family planning satisfied among all WRA [7,9]. The projections can be used in policy and programme planning. For example, probabilistic projections can be used to set ambitious yet achievable country-level targets, and the model could be applied to subnational data [30,31]. Our results would allow this work to be extended to all women.

Furthermore, the results of our study are needed to underpin models of the impact of increased use of family planning on other reproductive health outcomes [13,32] and models estimating the wider impact on health, schooling, and economic outcomes. Worldwide, an estimated 44% of pregnancies were unintended in 2010–2014, and some 56% of all unintended pregnancies ended in abortion [33] these findings underscore the continuing need for investments to meet women's and couples' contraceptive needs. Globally, UWRA account for a significant share of unintended pregnancies and abortions, and these events could potentially lead to more serious consequences for UWRA than for MWRA. The annual number of abortions worldwide is estimated at 56.3 million in 2010–2014, of which 27% were obtained by UWRA [34]. Both studies [33,34] pointed out that the lack of estimates of contraceptive use and unmet need for family planning among UWRA limited their ability to determine the extent to which these trends were associated with need for contraception. Our study will help close this gap.

The findings support recent calls to increase investments in family planning, especially in regions of the world where contraceptive prevalence is still low, unmet need is high, and the growth in the number of WRA is rapid. We show that more than half of the women with demand for family planning in 2030 will be in low-income and lower-middle–income countries, making it particularly important for the international community to support family-planning programmes here.

## Supporting information

**S1 Appendix. Technical appendix providing supplementary information on input data and methodology, plus supplementary results.**
(PDF)

**S2 Appendix. Supplementary figures; country-specific estimates and projections of family-planning indicators.**
(PDF)

**S1 Data. Input data for MWRA.**
(CSV)

**S2 Data. Input data for UWRA.**
(CSV)

**S3 Data. Table G in S1 Appendix provided in csv format.**
(CSV)

**S1 Results. Tables 1 and 2 in xlsx format.**
(XLSX)

**S2 Results. Table H in S1 Appendix provided in csv format.**
(CSV)

**S3 Results. Table I in S1 Appendix provided in csv format.**
(CSV)

**S4 Results. Table J in S1 Appendix provided in csv format.**
(CSV)

**S5 Results. Table K in S1 Appendix provided in csv format.**
(CSV)

**S6 Results. Table L in S1 Appendix provided in csv format.**
(CSV)

**S7 Results. Table M in S1 Appendix provided in csv format.**
(CSV)

**S1 Checklist. Guidelines for Accurate and Transparent Health Estimates Reporting (GATHER) checklist.**
(DOCX)

## Acknowledgments

Disclosure: The views expressed herein are those of the authors and do not necessarily reflect the views of the United Nations.

The authors gratefully acknowledge Ann Biddlecom, Jorge Bravo, Patrick Gerland, and John R. Wilmoth for their helpful comments on the paper. In addition, we thank Helena Cruz Castanheira, Stephen Kisambira, Kyaw-Kyaw Lay, Natalie Lin, and Nadia Soerjanto for their help with compiling the datasets of key family-planning indicators and Leontine Alkema, Win Brown, Niamh Cahill, Emily Sonneveldt, John Stover, and Michelle Weinberger for their valuable contributions to discussions regarding model development.

## Author Contributions

**Conceptualization:** Vladimíra Kantorová, Mark C. Wheldon, Philipp Ueffing, Aisha N. Z. Dasgupta.

**Data curation:** Vladimíra Kantorová, Philipp Ueffing, Aisha N. Z. Dasgupta.

**Formal analysis:** Mark C. Wheldon, Philipp Ueffing, Aisha N. Z. Dasgupta.

**Funding acquisition:** Vladimíra Kantorová.

**Methodology:** Vladimíra Kantorová, Mark C. Wheldon.

**Project administration:** Vladimíra Kantorová.

**Software:** Mark C. Wheldon.

**Supervision:** Vladimíra Kantorová.

**Validation:** Mark C. Wheldon.

**Visualization:** Vladimíra Kantorová, Mark C. Wheldon, Philipp Ueffing.

**Writing – original draft:** Vladimíra Kantorová, Mark C. Wheldon, Philipp Ueffing, Aisha N. Z. Dasgupta.

**Writing – review & editing:** Vladimíra Kantorová, Mark C. Wheldon, Philipp Ueffing, Aisha N. Z. Dasgupta.

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
