## [Decision Letter · Decision Letter 0]

27 Oct 2019

Dear Dr. Kantorova,

Thank you very much for submitting your manuscript "Contraceptive prevalence, unmet need for family planning, and the need for family planning satisfied by modern contraceptive methods: National, regional and global trends for all women of reproductive age from 1990 to 2030" (PMEDICINE-D-19-03179) for consideration at PLOS Medicine. 

[LINK]

In light of these reviews, I am afraid that we will not be able to accept the manuscript for publication in the journal in its current form, but we would like to consider a revised version that addresses the reviewers' and editors' comments. Obviously we cannot make any decision about publication until we have seen the revised manuscript and your response, and we plan to seek re-review by one or more of the reviewers. 

We expect to receive your revised manuscript by Nov 08 2019 11:59PM. Please email us (plosmedicine@plos.org) if you have any questions or concerns.

We look forward to receiving your revised manuscript. 

Sincerely,

Louise Gaynor, MBBS PhD

Associate Editor 

PLOS Medicine

plosmedicine.org

Title – it’s quite long and we also need a study descriptor, can we instead change to: Progress in 2030 sustainable development goals for contraceptive access and use: predictive model estimates from 185 countries

Abstract – Please provide summary demographic details of women included in the survey; please use 95% Cis and p values here (and throughout including the main text and tables, etc) for any quantifiable data.

At this stage, we ask that you include a short, non-technical Author Summary of your research to make findings accessible to a wide audience that includes both scientists and non-scientists. The Author Summary should immediately follow the Abstract in your revised manuscript. This text is subject to editorial change and should be distinct from the scientific abstract. Please

see our author guidelines for more information: https://journals.plos.org/plosmedicine/s/revising-your-manuscript#loc-author-summary

Please avoid using itals font for emphasis

Refs in the main text, please use square brackets instead of superscript and the brackets and refs should occur before the full stop. 

Did your study have a prospective protocol or analysis plan? Please state this (either way) early in the Methods section.

Please ensure the doc is clean and clear of track changes – page 8.

Please clarify how many countries are included – abstract you say 185, under the ‘Data compilation’ section you say 195. 

Please clarify in the main text what traditional and modern contraceptives are, with examples for relevance. Please place this ahead of the first mention in table 1. 

Fig 4 – more developed / less developed countries. This is rather vague. Is this common terminology? If not please use something more specific.

Please provide a completed GATHER checklist as a Supp file. 

Comments from the reviewers:

Reviewer #1: The paper is well written and organized and addresses and important global health issue. It will likely be widely cited. Past studies of FP indicators (eg. by the UN) have often focused on married/in union women, thus neglecting a growing population of unmarried women. The present study provides for the first time estimates and projections of FP indicators by country and region for both married and unmarried women.

There are no issues that are problematic or require major revision. However I have several suggestions for clarifying the presentation.

1) The abstract and introduction should state clearly that you will present estimates from 1990 to 2019 and projections from 2019 to 2030

2) Introduction: between the first and second sentence add something like "past estimates of FP indicators have been largely limited to married/in union women but recent changes in ….."

3) To provide the reader with a better understanding of the importance of unmet need and unsatisfied demand, note that there x million abortions and Y million unplanned pregnancies annually with wide ranging adverse consequences.

4)The term "posterior" (used from page 8 onwards) should be defined for readers who are not familiar with Baysian methods

5) The PPI measure should be defined and the reader should be given some guidance as to how to interpret a value of say 0.9.

6) p13 line 5 and 6 says: " ...are largely a reflection of differences in sexual activity among UWRA" This statement is probably correct but how do you determine this?

7) Page 23 first para: Isn't a rise in sexual activity among UWRA part of the explanation?

Reviewer #2: Monitoring progress towards universal access to family planning, and associated indicators has been identified as a global priority. Unmarried women are often excluded from studies and demographic health surveys, which biases reporting of key family planning indicators. The authors present a model to help quantify the contribution of unmarried adolescents to these FP indicators, which is helpful for programs and policymakers. However, the authors miss a few opportunities to be more transparent in the main text on how the indicators were derived for unmarried women, particularly sexual activity which is used to create the indicators, and the sources of information that were used to make this estimate. In addition, the authors could make more comparisons to put their results in context. For example, some countries have included unmarried adolescents in their DHS surveys. How do the results compare for these specific countries. Also, it is unclear what the intention of including the married women indicators are, and how these indicators are different than prior estimates. It would be plausible to compare the unmarried/married, and also the total (All women) to show the impact of excluding the unmarried women is. Overall the analysis is interesting, and transparency in key methods in the primary text (not appendices) would improve comprehension about the methods used to create the model. Finally, the article is quite long and it would be helpful for the authors to more briefly describe the methods (omitting some of the modeling jargon but adding details critical to understanding how estimates were derived), cut the text to highlight key results and make more comparisons between unmarried/married/all women, and synthesize results more concisely in the discussion. These edits would greatly improve the manuscript. 

Specific comments are included below:

Abstract 

1. The conclusions state the CPR and demand for FP differ by marital status yet only unmarried women are described in findings. Add in findings from married women.

2. The last sentence of the conclusion is essentially same as the background. The data do not clearly indicate why "accurate monitoring" is essential.

Introduction 

3. The last sentence could be shortened to more concisely describe what the objective was and the other details can only be mentioned in methods.

4. More details on unmet need for UWRA who have not initiated sexual activity should be briefly discussed as you include this specific group in your methods.

Methods

5. Please specify the duration of delaying future childbearing in your definition. Also be clear that when you say delay you mean future children, not necessarily delay first pregnancy.

6. Omit the rational "caution" in the methods. "While FP indicators…"

7. It would be helpful if definitions were applied universally when indicated, ie at risk of pregnancy, rather than described for one group and referred to as being calculated in the same way.

8. Suggest re-defining calculation of unmet need rather than just citation.

9. It would be helpful to briefly describe how you determined if "social values shows most of the population regarded sexual activity among unmarried as not acceptable/justifiable". This is a critical component of parameterizing your model and should be described in methods not just in appendix.

10. The methods would benefit from more specificity rather than general approaches. Ie "estimates for countries and indicators with no data….", which countries or how many where there? Which indicators? More transparency in the methods would improve readability in the methods, yet also ensure description of methods is appropriate for the journal audience (most of whom aren't modelers/statisticians). 

11. How did the authors determine if the model was "robust and adequately calibrated"? Be more specific.

12. Define traditional method use.

13. I suggest adding prior published estimates of indicators on graphs where available, as data points or lines. This would improve interpretation of how the model for married women compares with prior estimates, as it is unclear what added value the model has for calculated indicators for married women rather than as a comparison using similar methodology for unmarried women.

Results

14. The authors begin by summarizing the changes in FP indicates among married women over time and geographically. However the primary objective is to show unmarried women, this should be the focus of the results and comparisons made with married women.

15. Figure 1 shows data through 2030 but the text refers to 2019. Suggest harmonizing presentation to include 2030 for consistency, even though it is projected.

16. Results could be presented more succinctly to highlight key points rather than more comprehensive text on results (ie perhaps not all geographic regions need to be mentioned in text for all indicators).

17. Presentation of married/unmarried women separately in the text makes it more difficult to compare findings. I suggest reorganizing to pair the indicators and present married/unmarried together.

18. Figure 4, need legend to understand income group. Also, suggest moving to appendix.

19. Avoid interpreting results in results section, ie interpreting the explanation why total need for FP increases slightly. It is also redundant with the discussion.

20. The comparison of indicates with Guttmacher is helpful; there appear to be a few nuances of the data prior to or including 2017. Could you do a sensitivity analysis and compare your 2016 estimates to their 2017 to help account for this difference?

Reviewer #3: 

This is an interesting paper concerned with unmet need for family planning. It is well written and represents the result of many laborious tasks. The methodological aspects of the work are well selected and suitable for the applied problem.

Specific comments are given below.

Linking the evidence on MWRA and UWRA is a vital aspect of the work. The way this is performed is described in the appendix where the details of the hierarchical model which allows for borrowing of strength are outlined. However, it would be illuminating if the principles of this aspect are also given in the main manuscript, especially the intuition of how is this crucial task accomplished, including the flow of information.

It would be useful to add the priors for the (co)variance matrices (like in 3.47), especially since their density and parameters may affect the results, particularly in hierarchical models with borrowing of information and sparse data, as in the present paper.

Also, please discuss the choice of priors when restricted ranges are employed and informative selections are made, like in 3.120, say.

Please discuss the selection of two (high/low) sexual activity categories. Would a third (say intermediate) category be feasible in terms of data availability? Would this be expected to improve the accuracy of the results?

Please add the process of reviewing the evidence which resulted in the data collection. Was that done via a systematic review? Add the details where necessary, including the databases searched and the keywords used

[LINK]

---

## [Decision Letter · Decision Letter 1]

10 Dec 2019

Dear Dr. Wheldon,

Thank you very much for re-submitting your manuscript "Progress towards meeting contraceptive needs in the SDG era: Bayesian hierarchical model estimates and projections for 185 countries" (PMEDICINE-D-19-03179R1) for review by PLOS Medicine.

I have discussed the paper with my colleagues and the academic editor and it was also seen again by xxx reviewers. I am pleased to say that provided the remaining editorial and production issues are dealt with we are planning to accept the paper for publication in the journal.

[LINK]

We look forward to receiving the revised manuscript by Dec 17 2019 11:59PM. 

Sincerely,

Louise Gaynor-Brook, MBBS PhD

Associate Editor 

PLOS Medicine

plosmedicine.org

Requests from Editors:

Title: The title is still rather long. We suggest “"Estimating progress towards meeting women's contraceptive needs in 185 countries from 1990-2019: a Bayesian hierarchical modelling study" or similar.

Abstract: Background - The final sentence should clearly state the study question. 

Please avoid assertions of primacy ("This study is the first..."); we suggest adding ‘To our knowledge...’ 

Please begin your Abstract: Conclusions with "In this study, we observed ..." or similar 

Please add the title ‘Author Summary’ ahead of your author summary

At the start of your methods section, please adapt the text to "In this study, contraceptive prevalence was defined ..." or similar. 

At the top of p.11 of the PDF, you state that "a protocol or analysis plan was not pre-published". Please adapt this wording to state whether or not a prespecified plan was prepared, and if so attach the relevant document as a supplementary file. Please highlight non-prespecified analyses. 

Please add a brief statement to your methods section to note (for example) that ethics approval was not required for this study. 

To begin the first paragraph of your discussion section, please add "Our findings indicate that ..." or similar. 

Please revise any references in superscript and present in square brackets e.g. line 284

Line 322 - Please add a space between ‘changewith’

Line 334 - please revise to ‘Estimates from an additional’

Fig 1a, 1b, 4 - please define GNI in the figure legend

Line 463 - please revise to ‘Table 1’

Line 513 - please revise to ‘The majority...’

References - please ensure that references are appropriately formatted e.g. removal of ‘The ‘ before ‘Lancet’ in ref 8 and others 

Please remove quotation marks from reference titles, e.g., reference 31. 

Where necessary (e.g., reference 33), please ensure that 6 author names are listed, followed by "et al.". 

Please adapt the format of reference 24 to match the other references.

Comments from Reviewers:

Reviewer #2: The revised manuscript has improved the flow and clarity. Only one comment on the revision:

Abstract - while I understand the rationale for not including married women in results of abstract, the conclusion of the abstract is based on this comparison of unmarried/married. I suggest revising your conclusion, "Trends in contraceptive prevalence and demand for family planning differ substantially by marital status" or showing the results on married women in the abstract. Otherwise your primary conclusion is not substantiated with evidence in abstract.

Reviewer #3: The authors have sufficiently revised an already very good paper so this article is now acceptable for publication

[LINK]

---

## [Editor Report · Decision Letter 2]

9 Jan 2020

Dear Dr Kantorová, 

On behalf of my colleagues and the academic editor, Dr. Alison L Drake, I am delighted to inform you that your manuscript entitled "Estimating progress towards meeting women's contraceptive needs in 185 countries: a Bayesian hierarchical modelling study" (PMEDICINE-D-19-03179R2) has been accepted for publication in PLOS Medicine. 

PRODUCTION PROCESS

PRESS

PROFILE INFORMATION

Thank you again for submitting the manuscript to PLOS Medicine. We look forward to publishing it. 

Best wishes, 

Louise Gaynor-Brook, MBBS PhD

Associate Editor 

PLOS Medicine

plosmedicine.org